# A Survey on Challenges and Progresses in Blockchain Technologies: A Performance and Security Perspective

**Xiaoying Zheng [1], Yongxin Zhu [1,\*] and Xueming Si [2]**

[1] Shanghai Advanced Research Institute, Chinese Academy of Sciences, 99 Haike Road, Shanghai 201210, China; zhengxy@sari.ac.cn

[2] School of Computer Science and Technology, Fudan University, 825 Zhangheng Road, Shanghai 201203, China; sxm@fudan.edu.cn

\* Correspondence: zhuyongxin@sari.ac.cn

**Abstract:** Blockchain naturally fits multiple industry sectors due its characteristics of decentralization, enhanced security, tamper-proof, improved traceability and transparency. However, there is a significant concern of blockchain's performance, since blockchain trades off its performance for a completely distributed feature, which enhances its security. In this paper, we investigate the state-of-the-art progress of blockchain, mainly from a performance and security perspective. We extracted 42 primary papers from major scientific databases and 34 online technical articles. The objective is to understand the current research trends, challenges and future directions. We briefly introduce the key technologies of blockchain including distributed ledger, cryptography, consensus, smart contracts and benchmarks. We next summarize the performance and security concerns raised in the investigation. We discuss the architectural choices, performance metrics, database management enhancements, and hybrid blockchains, and try to identify the effort that the state-of-the-art has made to balance between the performance and security. We also make experiments on Ethereum and survey other popular blockchain platforms on the scalability feature of blockchain. We later discuss the potential applications and present the lessons learned. Finally, we attempt to identify the open issues and possible research directions.

**Keywords:** blockchain; distributed ledger; consensus; smart contracts; decentralization

## 1. Introduction

The blockchain is one of the technologies that emerged in the latest decade and attracts great attention from both the industry and academia. A blockchain is a growing list of records (referred to as blocks) that are linked using cryptography. Each block contains a cryptographic hash of the previous block, a timestamp, and the transaction data that are represented as a Merkle tree. A blockchain is well known for the feature of data immutability. It is in fact an open and distributed ledger running over a peer-to-peer (P2P) network that can manage transactions for multiple entities efficiently without a middleman and in a verifiable and traceable way. The tamperproof nature of blockchain comes from the fact that, once consensus is reached and a block is committed, the data in the block cannot be altered retroactively without alteration of all subsequent blocks, which requires consensus of the majority. Blockchain was first invented by an anonym named Satoshi Nakamoto in the milestone paper of Bitcoin in 2008 [1]. Bitcoin is the first digital currency that can detect the double-spending problem without the interference of a trusted authority. Since then, many other blockchain platforms (Ethereum, Hyperledger Fabric, Ripple, and Litecoin, etc.) and applications not all limited to finance have been inspired by Bitcoin, and blockchain has become one of the hottest and most promising technologies in recent years.

Although blockchain shows a huge prospective in the coming future, some performance and security concerns have to be considered and solved. For instance, the DAO, the distributed autonomous organization was hacked on 18 June 2016. A total of 3.6 million ether was drained into a child DAO that has the same structure as the DAO, which damages the reputation of Ethereum and raises the public's suspicion on blockchain [2]. Another concern is that transacting in Bitcoin is painfully slow. A theoretical maximum speed for Bitcoin that has been circulating online is 7 tps (transactions per second), and the achievable throughput in reality is only 3–4 tps. Meanwhile, Paypal achieves 193 tps and Visa achieves 1667 tps on average. It is difficult to directly compare centralized system, such as Visa, with decentralized ones, such as Bitcoin, but the fundamental problem of scalability keeps the blockchains operating in beta and alpha mode in business and industry. All these security and performance concerns are due to what is called the "Scalability Trilemma", where a blockchain system tries to offer scalability, decentralization and security, without compromising any of them. Decentralization is the core property that enables the censorship-resistance and permissionless features. Scalability is the ability to process transactions on a network with increased size. Security is an important component that guarantees the immutability of the ledger and its resistance to general cyber attacks. Unfortunately, it is believed that, at a fundamental level, blockchain can only simultaneously guarantee two out of three of these traits at one time. In general, public blockchain platforms are designed with a focus on decentralization and security, which undoubtedly leads a compromise in the scalability and an incredibly low transaction rate. As the blockchain technology is young, there are many solutions still in development, or currently in the market, that attempt to solve the scalability problem. In this survey, we explore various alternative blockchain architecture options that make different three-way trade-offs among decentralization, scalability and security.

There are several good survey papers on blockchain, and our scope and aims in this paper are different as we investigate the challenges and research trend from the performance and security perspective of blockchain. Some survey papers focus on the specific aspects of blockchain such as security [3], privacy [4], architectures [5], consensus protocols [6,7], smart contracts [8], applications, and Internet of Thing (IoT) security related applications [9–11]. Some papers focus on surveying the security and privacy issues of specific blockchain platforms, such as Bitcoin [12] and Ethereum [13]. There is also a systematic survey on 41 highly-selected blockchain related papers [14], and the goal is to analyze the current research trend in blockchain area. We focus on the security and performance aspects of blockchain, discuss the major performance bottlenecks and the potential security breaches, identify the trade-off between performance and security, and examine open challenges for future development and research. We also briefly cover the main concepts of blockchain and introduce the recent progress in blockchain related industry domain.

The rest of this paper is organized as follows. We introduce the general structure and key technology components of blockchain in Section 2, including the distributed ledger, cryptograph, the consensus protocols, smart contracts and the benchmarks. We next discuss the state of art progresses and challenges from the performance and security aspects in Section 3. Section 4 discusses blockchain industrial applications in different domains. Section 5 summarizes the lessons learned about performance and security, and provides the research trend and open issues in the field. Section 6 concludes the paper.

## 2. Key Technologies

Blockchain is originally linked blocks distributed over multiple peer nodes and deployed in an untrusted environment. Each block contains the transaction data, the hash of the previous block, and its own hash, as shown in Figure 1.

The survey work in [11] describes different components in blockchain systems, as shown in Figure 2. The distributed ledger is the database of the blockchain that stores data and is designed to be tamper-proof. A peer network carries an identical copy of the ledger on each of its peers, runs the protocol and enables the blockchain transactions without an authority. Membership services provide

the function of user authentication and management. A smart contract is a program allowed to run in a blockchain platform and perform specific functions. Cryptography is used to ensure the integrity of data. Events are update notifications that happen over the network, including new transactions, creation of new blocks, the arrival of new peers, etc. System management provides the function of creation, monitoring, and modification of components in the system. Finally, the component of system integration is used to integrate the blockchain system with external systems. Next, we introduce the key technologies and mechanisms of these components in the blockchain architecture.

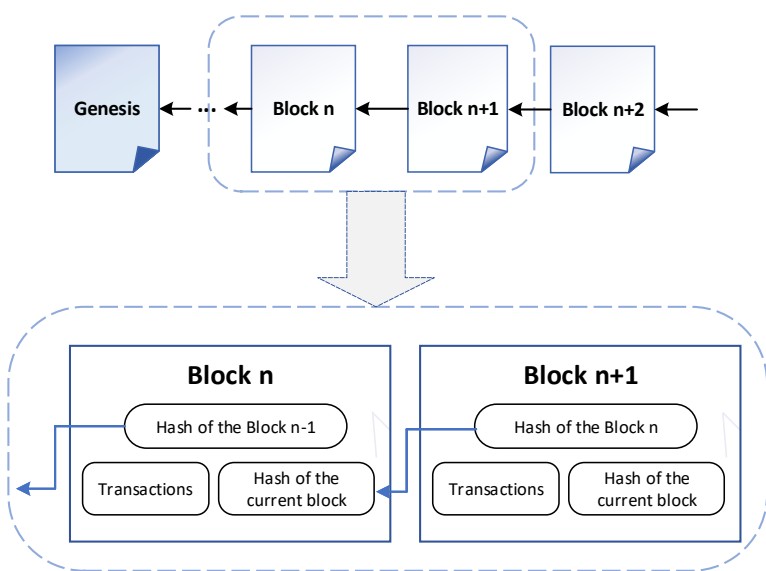

**Figure 1.** The chain data structure of blockchain.

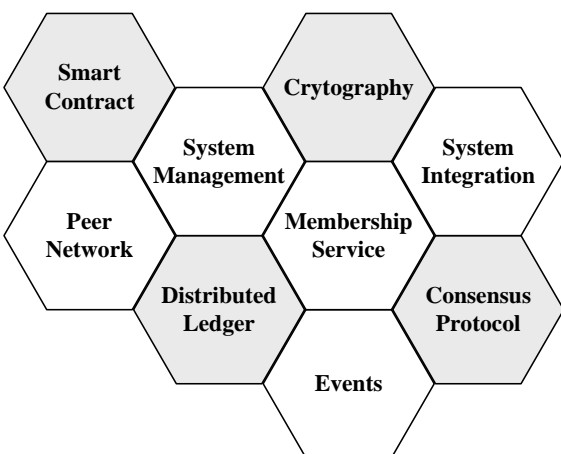

**Figure 2.** Major components in blockchain systems.

## 2.1. Distributed Ledger

A distributed ledger is a data storage spread across multiple nodes over a peer-to-peer network, where each node holds a complete and identical copy of the data. Since there is no central administrator in a distributed ledger, the data consistency is maintained by consensus algorithms to ensure the integrity of the data replicas on each independent node. More specifically, a distributed ledger is an append-only data storage, and the transaction update operations are recorded at each individual node in an un-coordinated fashion. Multiple nodes must run a consensus algorithm to reach an agreement at the commit order of transactions, and thus, reach a consistent state agreed on by all parties.

## 2.2. Cryptography

In a blockchain system, data integrity is an essential requirement since blockchain is originally proposed and developed in an untrusted environment and any data tampering must be detected. Cryptography is the key in blockchain's effort to ensure the integrity of the distributed ledger. Various cryptography techniques such as hash, public key cryptography and zero-knowledge proofs of knowledge are widely used in blockchain.

The data integrity is protected by the hash (Merkle) tree and hash pointers together. In blockchain systems, only the current global states are maintained, and the history of past states can only be achieved by walking through the block transaction history. The current states are protected by the Merkle tree. In a Merkle tree, each leaf contains the system states; each non-leaf node contains the hash of its child nodes. Thus, the root contains the root hash of the overall ledger and any state change will result in a new root hash. Hence, Merkel trees allow efficient and secure verification of the distributed ledger. On the other hand, the block transaction history is protected by hash pointers. Since the distributed ledger is an append-only data storage, a block is immutable once it is committed to the blockchain. To ensure a block cannot be altered once it is appended to the ledger, a block with an index of $n + 1$ contains the hash of the predecessor block, i.e., block $n$. Thus, any change in block $n$ will result in the invalidation of all following blocks. By combining Merkel tree and hash pointers together, the data integrity is ensured.

## 2.3. Consensus

A blockchain is completely decentralized over multiple nodes without a central coordinator. Therefore, to keep the consistency of the distributed ledger, a blockchain needs to come to a consensus on the block transaction history. Consensus is a dynamic way of reaching an agreement in a group. As blockchains can be exposed to a malicious environment, consensus algorithms are well known for suffering from the Byzantine Generals problem. Several consensus algorithms have been proposed and adopted in blockchain systems which can solve the Byzantine Generals problem. We describe the major consensus algorithms and list the comparison in Table 1.

**Table 1.** Comparison of the major consensus algorithms [3].

| Features | PoW | PoS | DPoS | Raft | pBFT |
|---|---|---|---|---|---|
| BFT | 50% | 50% | 50% | N/A | 33% |
| CFT | 50% | 50% | 50% | 50% | 33% |
| verification speed | >100 s | <100 s | <100 s | <10 s | <10 s |
| tps | <100 | <1000 | <1000 | >10 k | <2000 |
| scalability | strong | strong | strong | weak | weak |
| typical platforms | Bitcoin | Ethereum [15] | BitShares [16] | Quorum [17] | Hyperledger Fabric [18] |

Bitcoin bypasses the problem by inventing the proof of work protocol (PoW). In PoW protocol, a cryptographic puzzle has to be solved in order to add a block to the blockchain, which requires immense amount of energy and computational usage and hence ensures avoiding the Byzantine attack. However, there are some well-known issues with PoW. First, PoW is an extremely inefficient process because of the paramount amount of power consumed. Second, PoW is not as decentralized as it hopes to be since nodes with a more powerful computing facility usually have better chances of solving puzzles.

Ethereum is considering changing its distributed consensus system to a more efficient protocol, the proof of stake (PoS). In PoS protocol, if a node wants to add a block to the chain, it will validate the block by placing a bet on it. After the block gets appended, the node will get a reward proportionate to its bet. Apparently, the PoS protocol is more resource-friendly than PoW. However, the PoS protocol suffers from the "Nothing at Stake" problem. Suppose there is a short chain branches from the main chain. In the PoW protocol, all nodes will continue to work on the longer chain because it is only a

waste of computing energy working on the shorter chain. In the PoS protocol, a malicious miner can mine on the shorter chain because there is no risk of losing any stake. Therefore, the PoS protocol cannot avoid chain splits.

BitShares shifts to an improved version of PoS, the delegated proof-of-stake (DPoS) consensus. DPoS consensus relies on the power of stakeholder approval voting, where the stakeholders can elect any number of witnesses and blocks are generated by elected witnesses in a sequential order without a global agreement. Thus, DPoS consensus is faster, more efficient and more flexible compared with PoS. The transaction confirmation only costs an average of 1 s. However, as DPoS trades the property of decentralization for efficiency, there are concerns that a small number of witnesses will lead to a very high degree of centralization and also that it faces real attacks such as shut down by governments or ISPs.

Paxos is a family of consensus protocols that make various trade-offs between assumptions about the processors, participants, and messages in a given system. The protocol guarantees safety, only considers the crash fault tolerance (CFT), and is often employed where the durability of large datasets is required. Raft is an understandable alternative of Paxos, and is equivalent to Paxos in fault-tolerance and performance. Neither Paxos nor Raft considers the malicious environment and only fit the private or federated blockchains such as Quorum.

Hyperledger Fabric adopts the practical Byzantine fault tolerance (practical BFT, i.e., pBFT) protocol as its consensus mechanism [19]. The pBFT protocol can work in a malicious environment that no more than $\frac{1}{3}$ nodes are dishonest. All of the nodes communicate with each other, and the goal is that all honest nodes help in reaching a consensus regarding the state of the system through the majority. The important advantage of the pBFT protocol compared to PoW is its significant reduction in energy consumption. However, the heavy communication overhead makes the classical pBFT only work with small consensus group sizes and have a poor scalability.

The work in [7] reviews the consensus protocols of private/federated blockchain platforms including Hyperledger Fabric, Tendermint [20], Symbiont [21], R3 Corda [22], Iroha [23], Kadena [24], Quorum, Sawtooth Lake [25], Ripple [26], Stellar [27], and IOTA [28]. The paper discusses and compares the fault models and resilience against attacks of the respect platforms. The work in [6] surveys and compares consensus mechanisms of P2P system and blockchains. Both P2P and blockchains are decentralized systems without an authority, but P2P systems can handle the security issue under a malicious and anonymous environment with a very large population. The authors concluded that P2P systems can provide guidance for blockchains to improve consensus and incentive mechanisms.

*2.4. Smart Contracts*

Smart contracts are computer codes stored and replicated in the blockchain system, and can result in ledger actions such as money transfer and product or service delivery. Smart contracts help a transaction automatically executed in a transparent, conflict-free, undeniable, faster and more secure way without the aid of a third party.

Some blockchain platforms only provide a limited set of templates that can be used to write contract scripts such as Bitcoin. Some other platforms can support a more comprehensive set of codes. Ethereum provides a Turing complete code for specifying arbitrary computation. Some platforms allow smart contract codes run in their native runtimes, while other platforms create virtual machines for executing contract codes. For example, Hyperledger Fabric relies on Dockers to run codes.

However, smart contracts still suffer from security and legitimate issues when they are exposed in an untrusted system and they lack the regulation from the government.

In the survey work of smart contracts [8], Wang et al. introduced the operating mechanism and typical platforms of blockchain-enabled smart contracts. They also proposed a research framework of smart contracts including a six-layer architecture which organizes smart contracts related research aspects. Finally, they summarized the technical and legal challenges, recent progresses, and possible application scenarios of smart contracts. Rouhani et al. reviewed the key concepts and recent research

progresses of smart contracts in [29]. The survey mainly includes the security methods, performance improvement approaches and potential applications of smart contracts.

*2.5. Benchmarks*

Performance benchmarks are very important features of blockchain platforms. There are no general tools and standards that can provide performance evaluations for different blockchain solutions. The Hyperledger group made a lot of effort in defining performance benchmark of blockchains. The Hyperledger performance and scale working group published a white paper that defines the platform-agnostic terms and key metrics in performance evaluation [30]. Hyperledger also developed Hyperledger Caliper, a benchmark tool for blockchain frameworks, which is able to integrate multiple blockchain solutions and provide an evaluation environment [31]. Hyperledger Caliper can report major performance indicators, including tps, transaction latency, resource utilization, etc.

Dinh et al. presented Blockbench, a general evaluation framework for analyzing private blockchains [32]. Blockbench provides APIs that can enable any private blockchain integrated to the evaluation framework, and real workload to evaluate the performance in terms of throughput, latency, scalability and fault tolerance. Dinh et al. further conducted evaluation on three major private blockchains including Ethereum, Parity [33] and Hyperledger Fabric.

## 3. Challenges and the State of the Art Progresses

Blockchain is a decentralized system and often deployed in an uncontrollable environment, and, thus, faces a great challenge from hackers and thieves who keep trying to fraud the transactions. On the other side, the extreme security and decentralized requirements of blockchain scarify the performance compared with the traditional transaction systems. Many new technologies have been explored recently to try to make a balance between security and performance. In this section, we discuss the state-of-the-art progresses and challenges faced by blockchain from various aspects.

*3.1. Performance Concern*

Blockchain is designed for use when there are no trusted parties and central administration, but immutability of data and security is important. Because of blockchain's inherently distributed and peer-to-peer nature, blockchain-based transactions can only complete when all parties update their respective ledgers, which can be a long process. Hence, the implementation of blockchain is expensive, and many of its functions are also provided by the traditional centralized data management solution, i.e., the rational database. However, blockchain's performance in terms of throughput is significantly degraded compared with the rational database. For instance, the throughput of Bitcoin and Ethereum is only 4 tps and 20 tps, respectively, while the average throughput of Visa and PayPal is 1667 tps and 193 tps, respectively [34]. The loss of performance is traded for an advantage of blockchain that it provides a reliable, robust and secure way to store data without any third party interfering. We summarize the difference between public blockchain and database management systems (DBMS) from various architectural aspects in Table 2.

**Table 2.** Comparison of public blockchain and DBMS.

|  | Public Blockchain | DBMS |
|---|---|---|
| Application | reliable data management between entities | efficient data management inside one entity |
| Consensus | Byzantine fault tolerance | crash fault tolerance |
| Centralized | decentralized | strong centralized |
| Network | P2P in a malicious environment | master-slave in a trusted enviroment |
| Admission Control | none | user and role management |
| Storage | immutable chain-based transactions | transaction logs |
| Data structure | Merkle tree | B tree |
| Administrator | none | super user |

Despite the strong security features of blockchain, it still needs great effort to improve the performance in order to make blockchain applicable in practice. The work in [35] investigates the performance of the pBFT consensus protocol under a large number of peers. It presents a model of stochastic reward nets (SRN) to evaluate the mean finish time of consensus process in pBFT. It also conducts a real blockchain network running a production-grade IoT application and validates the model with a small population of peers. Thus, the proposed SRN model can be used to estimate the bottleneck of pBFT with a larger number of peers. The single chain based architecture has been thought to be the bottleneck of blockchain since transactions cannot be processed in parallel and have to be committed one by one under the single chain scheme. The work in [36] presents a multiple chain based blockchain architecture, which allows a set of blockchains to run in parallel and makes the system more scalable and extendible. In the proposed architecture, there is a single main chain and a set of semi-independent sub-chains. There is a value swap layer that connects sub-chains and the main chain, and aids the interaction between sub-chains. The architecture also provides witness services to enable sub-chains transactions and cross chain transactions. Sanka and Cheung proposed to cache the blockchain data in the FPGA key value store in network interface controller (NIC) to improve the scalability and throughput of blockchain applications [37]. In the proposed solution, to improve the performance, all the hashed values are stored in a customized SHA-256 hash core and an actual hashing operation is performed only when the requested one is not a hashed value. The FGPA based solution has been implemented in the real Bitcoin core and the evaluation shows that it improves the throughput performance by 103 times when cache hits. Qi et al. proposed a cascade structure for blockchain addressing existing performance problems of blockchain. In the structure, multiple micro blocks storing transactions are inserted between any two key blocks, and key blocks only store the digest of micro blocks. Furthermore, the micro blocks are unordered, and transaction-repeat tolerated, which can speedup the block generation and improve the overall performance [38]. Eyal et al. presented Bitcoin-NG (Next Generation), a Byzantine fault tolerant blockchain protocol that shares the same trust model as Bitcoin [39]. Bitcoin-NG splits blockchain operation into two phases, namely leader election and transaction serialization, ensuring that the system is able to continually process transactions during leader election, and, thus, improving system performance to the limit bounded by the propagation delay of the network and the processing capacity of the individual nodes. Kan et al. attempted to replace the original chain structure with the graph data structure to improve the throughput of Bitcoin and Bitcoin-NG system [40]. The graph data structure allows parallel mining and benefits the transaction confirm time. Li et al. proposed a proof of vote (POV) based consensus protocol, where the agreement is reached by the agencies voting in the league without the depending on a third-party intermediary or uncontrollable public awareness. Compared with the fully decentralized POW protocol, POV has controllable security, convergence reliability and low transaction confirmation time [41].

### 3.2. Security Concern

Blockchain has some desirable features that would help to secure transaction data. These features include tamper resistance, distributed ledgers, cryptography secured records, and resilient to the single point failure. However, the maturity of this technology including security of the fundamental protocols and its applications is not enough, and more conditions and requirements need to be considered when blockchain is applied in business.

First, we discuss and summarize the difference between blockchain security and the standard cyber security [11].

- The goal is different. The goal of the standard cyber security is to eliminate external attackers from tampering the data. The goal of blockchain security is to make data tampering infeasible by storing data copies at as many possible locations.
- The infrastructure is different. The standard cyber system is usually centralized, and the security is achieved through the strict admission control and permission grant. The blockchain system is

concentrated on decentralization, and availability and integrity is provided by the distributed nature of the ledger.

- The environment is different. The standard cyber system is controlled by the security staff and a semi-trusted (if not fully trusted) environment is assumed. Blockchain is supposed to run on untrusted distributed devices without a central authority.

Anyway, despite the fundamental difference of blockchain and the standard cyber security, blockchain still faces the security challenges that bother the standard cyber system. The work in [11] discusses these challenges in detail. First, blockchains can be overloaded by DDoS (Distributed Denial of Service) attacks. Second, since all peers in the blockchain are equivalent, any defect in an endpoint can be a defect to the whole system. Third, the application code (i.e., the smart contract in blockchain) can be the security threat. Unlike the traditional system, smart contracts can be submitted by malicious users, and therefore any malicious code in a smart contract can lead to a severe consequence. Even if the blockchain system is theoretically tamper-proof due to the cryptographic nature and the consensus protocol, there still exists creative ways to cheat. For instance, a selfish miner can trick other nodes to waste time on already-solved crypto-puzzles, and therefore subvert a blockchain even if it has less than half of the power. A malicious user can take control of an honest node's communications and trick it to accept false blocks. Another suspicious guarantee of a blockchain system is that it is not as decentralized as it intends to be. The nodes are often dominated by peers with more mining capacity, which makes Bitcoin or Ethereum a centralized system in fact. In permissioned systems, to solve the security issues, the owner may be granted more authority, which can be thought to violate the fundamental nature of blockchain. Finally, integrating with existing systems, cryptographic key material management, and providing network connection with required QoS guarantees are thought to be the major security concerns by Hyperledger [42]. The concern arising from the network quality is due to the fact that most consensus algorithms are latency sensitive and one potential DoS attack can happen by disrupting communication.

Matsuo investigated the way to enhance the security of blockchains by the formal analysis and formal verification [43]. A framework is proposed to apply formal analysis to the implementation, protocol and language layer of blockchain by using existing standards and results. Henry et al. studied the blockchain access privacy in the work [4]. To ensure access privacy, anonymous communications networks, such as Tor (https://www.torproject.org), have been proposed as candidate solutions. The work in [4] challenges this approach by showing that privacy can be breached on the anonymous communication networks and new approaches have to be sought. The survey work in [12] reviews the known vulnerabilities and privacy issues in Bitcoin, and investigates the state-of-the-art security solutions.

*3.3. Architectural Choices behind Performance and Security Concern*

There are many architectural options for blockchain implementation, which make trade-offs among various performance aspects including security, complexity, scalability, transparency, privacy, throughput, and latency. We discuss these options in details.

3.3.1. Public vs. Private/Federated

Public blockchain, also called permissionless blockchain, allows anyone to participate without permission. The consensus protocols of public blockchain are often based on PoW, PoS or the hybrid. With public blockchain, anyone can run a public node on its own device, join the network, and participate in the transactions. Transactions in public blockchain are transparent but anonymous/pseudonymous, and can be read by any participant in the chain.

Private blockchain, also called permissioned blockchain, is used in a single organization to process transactions internally. In private blockchain, read permissions can be public or granted to a restricted group of participants, but write permissions are only granted to the organization. Private blockchain works in a trusted environment, and is much faster and cheaper than public blockchain because only

the owner keeps the write permissions and no consensus has to be reached. Thus, it is much less secure and closed as compared to public blockchain and suffers from the security breaches as that of a centralized system.

Federated blockchain or blockchain consortium is opposed to public blockchain and very similar to private blockchain, but there is a significant difference. Federated blockchain operates under the leadership of a group instead of a single organization, where multiple entities establish a decentralized system as that in public blockchain. Similar to private blockchain, write permissions are only granted to trusted entities and others are not allowed to participate in the process of verifying transactions. The read permissions can be public or restricted to the participants. Therefore, instead of only one organization being in charge of private blockchain, there are multiple entities working together to make the system more distributed. Thus, federated blockchain is faster and more scalable, and provides more transaction privacy compared with public blockchain; and it is more decentralized and secure compared with private blockchain. Federated blockchain is mostly used in the banking sector.

We summarize the comparison of public blockchain with private/federated blockchain in Table 3.

**Table 3.** Comparison of public blockchain with private/federated blockchain.

|                  | **Public**                          | **Private/Federated**                            |
| ---------------- | ----------------------------------- | ------------------------------------------------ |
| Access           | open read/write                     | permissioned read/write                          |
| Speed            | slow                                | fast                                             |
| Security         | PoW, PoS, other consensus protocols | PBFT, Raft, legal contracts, proof of authority  |
| Membership       | anonymous/ pseudonymous             | known identity                                   |
| Environment      | untrusted                           | trusted                                          |
| Typical platforms | BitCoin, Ethereum                  | Hyperledger                                      |

### 3.3.2. Byzantine vs. Non-Byzantine Consensus Fault Model

A Byzantine fault is a condition of distributed computing systems, where nodes may fail and malicious nodes may exist; and there is no knowledge about whether a node has failed or it is a malicious node. A consensus protocol is Byzantine fault tolerant if it can work under an environment with malicious nodes and general node failures, while a protocol is non-Byzantine fault tolerant if it only works for general node failures. Some consensus algorithms of blockchain are Byzantine fault tolerant, where honest nodes can still reach an agreement with the existence of malicious nodes. Some consensus algorithms are not designed to work under a malicious environment, and they assume all nodes are honest and can only tolerate general system failures.

Practical Byzantine fault tolerance is one of the Byzantine fault tolerant consensus protocols. It improves the original BFT consensus mechanisms, can tolerate $\frac{1}{3}(n-1)$ of malicious nodes, where $n$ is the number of peer nodes, and has a polynomial time complexity. It has been implemented and enhanced in some popular blockchain platforms such as Hyperledger Fabric.

Raft is an improved practical version of Paxos. It has the same performance and fault-tolerance as that of Paxos protocol. It is more understandable and addresses all major pieces needed for practical systems. It is not supposed to work under a malicious Byzantine environment, only considers general system failures, and can tolerate $\frac{1}{2}(n-1)$ of node failure. Hence, it is non-Byzantine fault tolerant and works for private blockchain.

Ripple protocol uses proof-of-correctness and requires all nodes to be identified. It is more efficient than PoW and PoS protocols, can tolerate $\frac{1}{5}(n-1)$ of malicious nodes and works only for permissioned blockchains.

PoW protocol is a probabilistic Byzantine agreement, which can tolerate a malicious environment when the malicious nodes' hashing power is strictly less than $\frac{1}{2}$, assuming high network synchronicity [44]. PoS protocol can also work in a malicious environment when the malicious nodes' stake is strictly less than $\frac{1}{2}$. Thus, both PoW and PoS protocols work for public and federated blockchains.

Lei et al. claimed that the equal right of each member in pBFT is not reasonable since fault nodes should have less rights [45]. They presented a reputation based Byzantine Fault Tolerance (RBFT) algorithm, which evaluates each node's contribution together with its reputation in the voting process. A faulty node will get lower reputation and hence less right in the consensus process, if any malicious behavior is detected. The RBFT algorithm also assigns more priority to nodes with higher reputation when validating new blocks. RBFT algorithm can tolerate $\frac{1}{3}(n-1)$ of malicious nodes as that of pBFT, but it has higher tps and lower latency compared with pBFT when the faulty node rate is high.

### 3.3.3. On-Chain vs. Off-Chain Data Storage

Intuitively, data stored on chain are more secure, trustable, tamper-evident, and single point of failure tolerant. However, data on-chain are available to everyone in the chain and with little privacy guaranteed. The amount of data stored on chain is limited, either because the limitation of the protocol or because of the huge cost of storing data on-chain. The storing cost is very high because every node in the blockchain has to keep a complete copy of the data. This is why even storing kilobytes can cost a fortune. Further, it is hard to query for data in blockchain because blockchain is not a SQL server and querying on-chain data is very expensive.

On the contrary, off-chain data provide privacy and have negligible storing cost, but they do not take advantage of the security, reliability and trustability features of blockchain.

There are many options for storing data on the blockchain. Instead of storing the complete raw data on-chain, we can choose to store only the hash or a subset of data on-chain. By storing only the hash or a subset of data, the storing cost is magically reduced. The raw data can be stored in a relational database or in a file system with the corresponding hash value used as the index.

### 3.3.4. Transactional Stores vs. Non-Transactional Stores

Blockchain often uses a transaction model to make a record. A transaction is a state transition that changes data in the blockchain from one value to another. These data can be any type of data tracked in the blockchain, including a cryptocurrency amount, IoT sensor readings, etc. A transaction includes the address of the sender, the address of the receiver, and the exchange between the two parties. To store the custom data on the blockchain, we need to package data into transactions to be able to store it. Some blockchain allows custom data to be appended to a transaction within their protocols, which are called transactional stores.

Some blockchain does not have that feature. Hence, custom data have to be encoded (if necessary) and used as a receiver's address. A transaction is sent to this specific address and the data are stored in the blockchain. By the non-transactional stores, data are encoded into the receiving address instead of storing in the payload field of the transaction. There are two drawbacks for the non-transactional stores. First, since the address size is limited, the number of data that can be stored is very small. Secondly, the transaction is sent to an address not owned by the sender; thus, the transaction fee is lost.

### 3.3.5. SQL vs. NoSQL Data Stores

On-chain storage has a limited capacity, thus we have to resort to off-chain mechanisms to store large amounts of data. There are two main options of databases, namely SQL and NoSQL, or relational databases and non-relational databases.

A relational database is the more rigid, structured way of storing data. The data stored in a relational database have to be structured in a very organized way. ACID (Atomicity, Consistency, Isolation, and Durability) compliancy is ensured by a relational database, which protects the integrity of data and reduces anomalies. Although a relational database is less scalable and does not support high traffic volume compared with a non-relational database, it fits for a more consistent dataset that does not grow rapidly. The popular relational databases include MySQL, Oracle, IMB DB2, Sybase, MS SQL Server, Microsoft Azure and PostgreSQL.

A non-relational database is document-oriented and distributed, which is more suitable for unstructured data from the web and can include sensor data, social sharing, personal settings, photos, location-based information, online activity, usage metrics, and more. The non-relational databases gain popularity for their speed and scalability, and fit for datasets that have large volumes and have little to no structure. There are four types of non-relational databases: key-value model, column store, document database, and graph database. MongoDB, Apache's CouchDB, HBase, Oracle NoSQL and Apache's Cassandra DB are the popular non-relational databases.

*3.4. Performance Metrics*

Because blockchain is implemented without a central authority, the system integrity and availability are enhanced by keeping redundant data across multiple nodes at some expense to performance. Thus, it is important to evaluate the performance metrics with the increasing node number and workload in blockchain. The Hyperledger performance and scale working group published a white paper that defines the platform-agnostic terms and key metrics in evaluating the performance of blockchain [30]. Dinh et al. presented a benchmarking framework for evaluating performance of private blockchain, and conducted a comprehensive evaluation of three major blockchain platforms including Ethereum, Parity, and Hyperledger Fabric [46]. In this section, we summarize the important performance metrics and present the performance comparison conclusion drawn by the research work [46] in Table 4. We only list Ethereum, Hyperledger and Parity in Table 4 to compare the performance. The reason is that the key performance metrics such as throughput and latency are only relevant when compared against a transaction of a similar nature on an identical set of constraints. The various blockchain technologies differ in how they define a transaction. For Ethereum, Hyperledger and Parity, the transaction is measured as the duration between the submission of a transaction to the network and the confirmation of the transaction to a block. For other blockchain platforms, the definition of a transaction might be very different. For instance, unlike the traditional blockchain, IOTA is a technology that is based on Tangle. IOTA uses a whole network, i.e., the tangle, instead of a chain to map transactions. Theoretically, IOTA will enjoy good throughput, latency and scalability since it trades the strict data consistency for performance improvement. Thus, it is difficult if not impossible to directly compare IOTA with other chain-based blockchain platforms in an identical environment. We also list IOTA in the performance comparison table, where the performance is evaluated with different methods or analyzed theoretically compared with the other three platforms.

- Throughput refers to the number of successful transactions that a blockchain platform can process per second.
- Latency is the response time per transaction of the blockchain platform.
- Scalability evaluates the changes in throughput and latency when the blockchain experiences the increase of nodes and concurrent workloads.
- Fault tolerance evaluates the changes in throughput and latency during node failure including system crashes, network delays and random message corruptions.
- Security metrics are defined as the ratio between the total number of blocks of the main branch and the total number of confirmed blocks. When the ratios is high, the blockchain system is vulnerable to the security attack resulted from double spending or selfish mining.

**Table 4.** Performance comparison of popular blockchain platforms.

| Metrics | Ethereum | Parity |
|---|---|---|
| Throughput | hundreds of tps | tens of tps (no more than 80 tps) |
| Latency | around 100 seconds | several seconds |
| Scalability | degrade linearly | keep constant |
| Fault tolerance | unaffected | unaffected |
| Security | vulnerable | vulnerable |
| | **Hyperledger** | **IOTA** |
| Throughput | thousands of tps | 127 tps on a 250-node network |
| Latency | tens of seconds | approach zero when the network reaches a certain size |
| Scalability | can not scale up to 16 nodes | scalability increases with more users |
| Fault tolerance | stop working after $\frac{1}{3}$ nodes fail | asynchronous BFT and works with more than $\frac{2}{3}$ honest nodes |
| Security | guaranteed safety | vulnerable |

Thakkar et al. conducted a comprehensive empirical study to characterize the performance of Hyperledger Fabric [47]. The study was conducted in an environment of eight peers and one orderer node with a kafka-zookeeper. All peers and kafka-zookeeper ran on the x86_64 virtual machines with 32 vCPUs and 32 GB of memory, and were connected to the 3 Gbps Datacenter network. The experiment increased the transaction arrival rate until it reached the saturation, and repeated the tests with varied block size. The work concludes that, with an increase in transaction arrival rate, the throughput increases linearly as expected until it reaches the maximum throughput around 140 tps. When the transaction arrival rate reaches the maximum throughput, the latency increases dramatically, more specifically from dozens or several hundred of milliseconds to hundreds of seconds. Furthermore, for an arrival rate less than the maximum throughput, the latency increases significantly as the block size increases; when the arrival rate reaches the maximum throughput, the latency decreases slightly as the block size increases.

We also tested the scalability of Ethereum in terms of the time and energy consumption with the increase of stored data. We ran the evaluation in the following environment. An Ethereum client Geth (1.8.3-stable) ran on a computer (MacBook Pro 2017 version) as well as an Ethereum Wallet 0.10.0. The hardware configuration was a macOS 10.14.6 operating system, a CPU with 2.3 GHz, an i5 version Intel Core and a memory with 8 GB of 2133 MHz LPDDR3. We explain the performance by testing gas cost and time consumption for documents stored on the blockchain. As shown in Figure 3, we found that, with the increase of documents, gas cost increases steadily. When the amount of documents is 10, the gas cost is 151,408. For additional 10 documents stored, 100,000 more gas could be cost. As shown in Figure 4, we found that, with the increase of documents, it shows an increasing trend. When the amount of documents is 10, the time cost is 952.001 s. When the amount of documents is 100, the time cost is 5636 s. The growth range of time cost is almost linear, which proves that the storage performance of Ethereum is high performance.

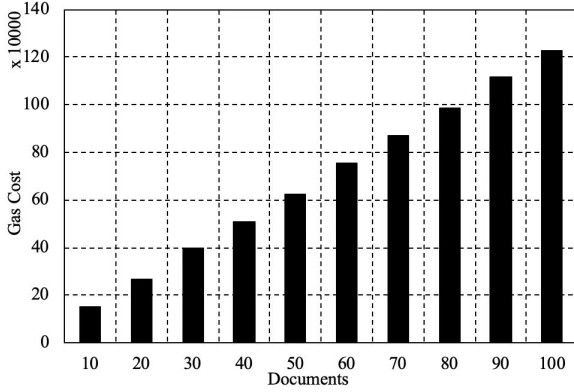

**Figure 3.** The gas cost with the increased documents in Ethereum.

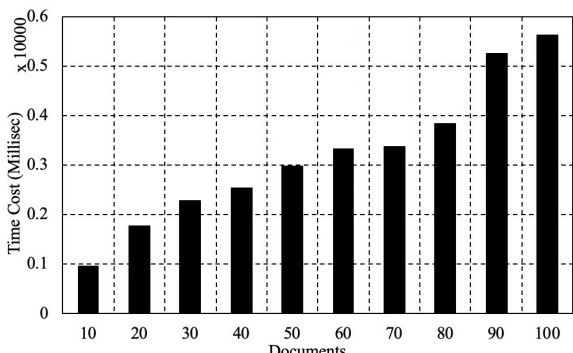

**Figure 4.** The time consumption with the increased documents in Ethereum.

## 3.5. DBMS Enhancements

It is possible to make a database decentralized, more secure, 100% trustworthy, transparent and more efficient by enhancing it with blockchain techniques. IBM has filed a patent to build a reliable database tampering detection system, which can detect any data inconsistencies in the central database by the blockchain technique. BigchainDB aims to provide a database with blockchain characteristics, which supports high throughput, low latency, powerful query functionality, decentralized control, immutable data storage and built-in asset support. BigchainDB version 2.0 makes significant improvements over previous versions, which is now Byzantine fault tolerant and can tolerate up to a third of node failures. The survey work [48] reviews the design goals of BigchainDB 2.0, the technical approaches, use cases and future plans. Corda is a decentralized global database upon a distributed ledger with minimal trust between nodes. It aims to serve various applications in finance, trade, supply chain tracking and more, and is specialized for use with regulated financial institutions [22,49]. Zhu et al. proposed a semantics empowered blockchain database, i.e., SEBDB, which considers both usability and scalability. In SEBDB, the off-chain data are stored in a rational DBMS and on-chain data are modeled as a number of predefined relations. Each transaction is a tuple with multiple attributes belonging to a certain relation and carries relational data semantics. The general interface is implemented with a SQL-like language instead of code-level APIs, which provides more convenience. Therefore, SEBDB enhances the blockchain platform by leveraging the existing databases' functionality which are optimized for decades [50].

## 3.6. Analytics on Chaincode Data

Blockchain provides more benefits to data analytics due to its security and redundant features. With blockchain, data cannot be forged and the fraud actions can be prevented in the real-time transactions. Combined with big data analytics, it is possible to identify behavior patterns and identify risky transactions in real-time. Blockchain also greatly improves transparency in data analytics, thus the behavior patterns identification is likely to be more accurate. Furthermore, with the redundant architecture of blockchain, data are more valuable and reliable.

On the other aspect, transactions are globally published and are not encrypted in most blockchain applications. Hence, there is raised concern of lacking privacy. One solution is to store only encrypted data in the blockchain, which leads to a security problem if the key to decrypt is lost.

Furthermore, for on-chain data, only the current states are recorded in the Merkel tree, and other transactional history in stored in the blocks which do not support fast query. Therefore, querying historical data is very time-consuming and expensive. In [51], Gupta et al. presented two models for efficiently processing temporal queries on Hyperledger Fabric. The first model creates a copy of each transaction event and stores temporally close transactions together on Fabric. The second model tags metadata to each transaction and makes temporally close events share the same metadata. In the follow-up work [52], Gupta et al. presented variants on the two models which can handle the skew in Fabric data.

### 3.7. Cross Channel Transactions

Inter-blockchain transactions are difficult because a trusted third party is required to execute inter-blockchain transactions and hence the benefits of blockchain are lost. Lightning Network has been actively working on a decentralized option for inter-blockchain transactions, which does not require a third party. The technology is called atomic cross-chain swaps, which allows two parties on two different blockchains to trade directly and instantly. Atomic swaps make use of a scheme known as a hashed timelock contract (HTLC), which combines two technologies, a hashlock and a timelock. A hashlock uses a cryptographic puzzle to ensure that one party cannot release their funds without the other doing the same. A timelock acts as a safety net if nothing happens, routing funds back to the senders after a certain amount of time. Currently, the technology of atomic swaps is still in its early stage. A user is required to download both the blockchains that the transaction involves before starting transactions. New development is expected to make atomic swaps more user friendly and further decentralize the market [53,54]. English et al. proposed a solution to the inter-blockchain transaction problem by encapsulating information regarding the fidelity of peer-to-peer transaction facilitators in a hierarchical meta-blockchain layer [55].

### 3.8. Hybrid Blockchains

The hybrid blockchain lies between the two extremes of public and private and attempts to use the best part of both private and public blockchain solutions. The hybrid blockchain will restrict the access, but still offers integrity, transparency, and security. The members of the hybrid blockchain can decide who can participate in the blockchain or which transactions are made public. The hybrid blockchain inherits the decentralized, secure, transparent and immutable features from the public one, but restricts the rights to view, modify and append/approve transactions to only selective members.

## 4. Applications

Blockchain technologies are expected to empower new innovations that either improve existing processes or unlock new technologies. In this section, we briefly review different industrial sectors where blockchain technologies has been proposed for application. We attempt to identify the security and performance requirements that support the implementation of blockchain for different industrial applications, and reveal the challenges that have to be addressed to achieve better utilization of this technology.

There are tremendous companies devoted to developing new applications based on blockchain technologies. Early in 2019, there were 197 blockchain service providers registered at the Cyberspace Administration of China [56]. The services provided can be grouped into nine categories as shown in Figure 5. Among these providers, about half provide a general blockchain platform or unrecognized blockchain services. The second largest category is financial service, which includes 43 providers and is obviously the most common application of blockchain. The following categories are supply chain services, copyright protection and legal services. There are also a few providers focusing on governance, healthcare, education and smart transportation. The survey work in [10] reviews different industrial application domains with blockchain technologies adopted, and explores the opportunities, benefits, requirements and challenges of incorporating blockchain in these applications.

**Blockchain Applications**

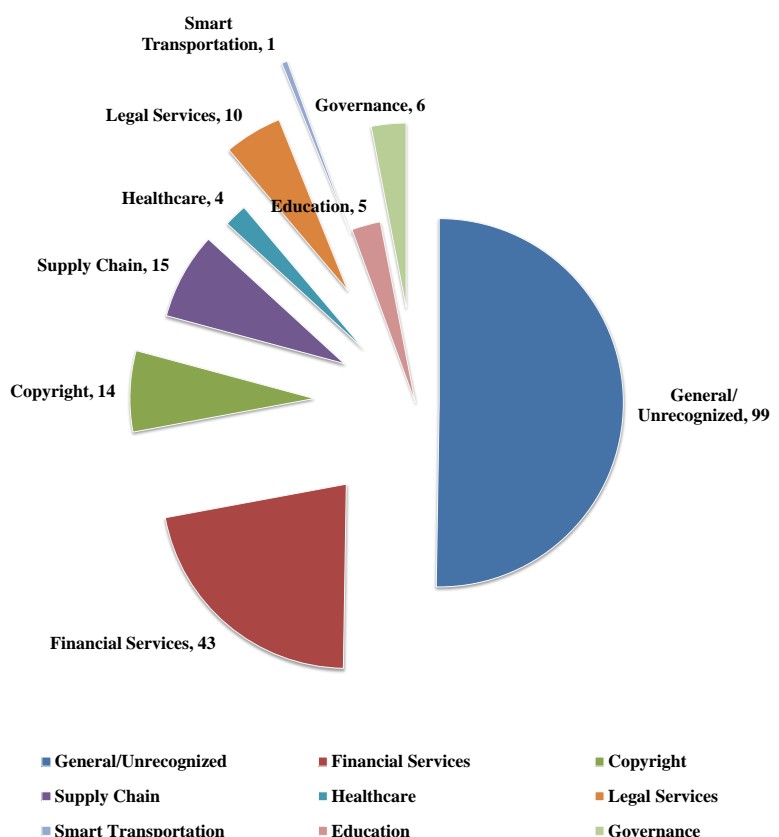

**Figure 5.** The statistics of blockchain service providers registered early in 2019 in China.

Below, we explore some of the most promising blockchain applications.

### 4.1. Financial Services

- Asset management: trade processing and settlement

  In traditional trade process, each party keeps their own records, which is expensive and risky. After switching to blockchain, the process can be simplified by distributed ledgers and errors can be reduced by encrypting the records. Furthermore, there is no need for intermediaries. Maersk, a Danish shipping company, was reported to aid the process of shipping flowers from the Kenyan port of Mombasa to Rotterdam by using a blockchain-based approach developed with IBM, where all documents were stored in the chain and shared by all participants [57].

- Insurance: claims processing

  Blockchains can provide a risk-free and transparent process for insurance claims, which are usually facing fragmented and unreliable data, and processed manually. State Farm, a mutual automobile insurance company in the US, announced that it is running a blockchain based test to streamline the subrogation process for auto claims through the first half of 2019. It is hoped that the blockchain solution can help to reduce the time and resource required to complete the subrogation process compared with a regular process [58]. The work in [59] presents a prototype of blockchain based fine-grained transportation insurance. In the prototype, information of drivers and vehicle usage are collected from mobile sensors and stored on an immutable, traceable, transparent

distributed ledger owned by different parties. Insurance premium are assessed based on the collected information of vehicles' usage and drivers' behavior, which promote fairness, encourage safer driving style and can be used by insurance companies for marketing.

- Payments: cross-border payments

  The cross-border payment takes at least days for money to cross the world, and is very costly and easily targeted by money laundering. Blockchain-aided remittance services can help to offer secure, cheap, instant and transparent international payments. In August 2016, SBI Ripple Asia announced that Ripple's blockchain technology will power payments and settlement for a Japanese bank consortium of 15 banks [60]. With the blockchain technology provided by Ripple, domestic and cross-border payments can be settled in real time and cross-border fees can be save up to 90%.

### 4.2. Internet-of-Things (IoT) Applications

In IoT applications, more and more intelligent devices are connected to the Internet and produce a large amount of data. Blockchain provides a reliable way to collect data from smart devices. The work in [61] designs a high performance blockchain platform to realize the decentralized autonomy of intelligent devices. Wang et al. used blockchain technology to enhance the security of IoT systems [62]. Roy et al. studied integrating blockchain in sensor networks to achieve security and privacy [9].

- Smart appliances

  A smart home appliance is a device that connects to the Internet and can help you to monitor and control the device when away from home. Encrypting these appliances on the blockchain can protect the ownership and enable transferability. Walmart has filed a patent application on the management of smart appliances using blockchain. The patent application proposes to manage a system of smart appliances such as computers, kiosks, tabletop devices, gaming devices, laptop computers, and portable media players using blockchain. It also describes a smart environment in a wide range of applications, such as home, media, manufacturing, energy, and healthcare [63]. Energy conservation measurements in buildings are measured as the difference between a predictive baseline model and the actual consumption, and the actual application involves a large amounts of data and often requires a third-party for auditing. The work in [64] develops a prototype for fair and trustable energy performance contracts, which stores the predictive models and the data in an immutable Ethereum based distributed ledger.

- Supply chain sensors

  Smart sensors provide companies the timely and valuable data of the supplies as they are transported, which can power insightful analysis that promotes improvements in cost, performance, or customer experience. With blockchain, smart sensors can be more secured, and any data-tamper can be tracked and located instantly. Walmart uses blockchain to keep track of sales of pork in China. Its system records where each piece of meat comes from, each processing and storage step, and sell-by-date in the chain [65]. The work in [66] attempts to find blockchain candidates that are able to ensure the integrity of supply chain by evaluating different blockchain alternatives.

### 4.3. Smart Contracts Applications

- Smart Grid

  A smart grid is an electrical grid which manages a variety of energy sources including renewable energy and energy measures. The advance of renewable energy and storage systems allows traditional energy consumers to sell excess energy back to the grid or other energy consumers within their vicinity, and hence makes the grid more distributed and in a peer-to-peer fashion.

Keeping tracks of the production and distribution of electricity is an important aspects of the smart grid. Blockchain technology can provide a secure and reliable way to carry peer-to-peer energy exchanges without a central coordinator based on smart contracts. This will lead to the increase in development of decentralized micro-grids and technologies enabling micro-grids. Gai et al. proposed a permissioned blockchain edge model for smart grid network to address the privacy protections and energy security issues in smart grid, by combining blockchain and edge computing techniques [67]. They used group signatures and covert channel authorization techniques to guarantee users' validity. There also exists a significant privacy issue in smart grid that energy trading volumes can be mined to detect its private information such as physical location and energy usage. In [68], a consortium blockchain-oriented approach is presented to solve the privacy leakage in smart grid and screen the distribution of energy sale of sellers.

- Blockchain Healthcare

  Blockchain can be used to store encoded personal health records and only allows granted parties to access. The distributed ledgers can also provide a secure and confidential health care management with the participation of the health care recipient, the health care provider, the insurance company, and the regulation agent. PokitDok, an e-commerce healthcare provider, introduces a blockchain based healthcare system named DokChain in 2016. DokChain integrates all endpoints in the healthcare ecosystem into the Sawtooth based blockchain network, and can provide health data for tracking and sharing in a secured way [69].

- Blockchain music

  In the online music industry, the centralized management takes profits from both artists and fans, where fans are charged huge fees for music access and artists give up much of their profits to the centralized management. The blockchain and smart contracts technology can help to create a distributed music streaming platform, where all transactions are transparent, the money go directly to the artists, and fans can access to more music in simpler and cheaper ways. There are many blockchain based music companies emerging worldly wide [70], spreading from the west coast of Europe to the east coast of China, and to the Americas. Ma et al. proposed a master–slave blockchain platform that is applicable in digital rights management [71].

### 4.4. Performance and Security Challenges Faced by Incorporating Blockchain in Industry

Now, we highlight the performance and security challenges faced by deploying blockchain in industrial sectors. First, the current application scales in industry are usually huge, which requires blockchain platforms to process transactions on a very high throughput rate. Most of the state-of-the-art blockchain platforms cannot handle these high transaction rates and the performance will be severely degraded. Especially for the financial services, the poor scalability and high transaction processing delays of blockchain become the major performance bottleneck. For IoT applications, the bottleneck is different. The work in [72] evaluates the performance of a cloud and edge hosted blockchain where blockchain is proposed as a service for IoT. The work shows that the network latency is the dominant performance factor when blockchain is adopted in IoT systems. Scalability has become a pressing issue with the rapid growth in the application size and transaction volumes. Secondly, applications in industry are deployed over the Internet, and hence are vulnerable for various cyber-attacks. Endpoint security is the major security concern in financial services. There are more challenges in linking the cryptographic identities to the real world identities, which can help to identify the suspects of money laundering. Many industrial applications such as healthcare applications involve sensitive or private information. Thus, it is important for blockchain technologies to provide different and acceptable levels of security and privacy measures that meet different needs of different industrial sectors.

## 5. Futuristic Topics

We present a comprehensive survey on the security and performance aspects of blockchain and the related techniques, and discuss some challenges and open issues in previous sections. We now summarize the lessons learned, and present the remaining challenges and the possible future research trend briefly.

### 5.1. Summary of Performance Limitations and Possible Directions

As discussed above, the performance of blockchain is severely degraded compared with the traditional database systems due to its inherently distributed, peer-to-peer nature. We summarize the major performance limitations that make blockchain inappropriate for many digital interactions as follows.

- Throughput

  The current throughput of commercial blockchain platforms is several magnitudes lower than the traditional database systems. With the expectation to apply blockchain to process business transactions in real production environment, the throughput of the blockchain network needs to be improved.

- Latency

  Currently, it takes dozens of minutes to complete a transaction in blockchain platforms, while a transaction only takes a few seconds in VISA. Thus, improving the processing latency to several minutes, while still maintaining security is a major challenge.

- Network bottleneck

  Since each node in blockchain network needs to keep a copy of data, the data exchanged in network makes the network bandwidth a bottleneck. As the size of a blockchain system increases, the network bandwidth bottleneck has to be solved.

We next clarify some performance related future research directions that might help making blockchain more practical for the real applications.

### 5.1.1. Parallelism Exploitation

The poor scalability and interconnection make it difficult for blockchain to bring about the large-scale commercial adoption. One reason is that the data structure of traditional blockchain is sequential. Blockchain can never scale up to a high tps since blocks have to be appended to the chain one by one in the sequential fashion. Improving the parallelism of blockchain may be a feasible approach. With a parallel data structure, there exists multiple chains and multiple blocks can be appended to the chains simultaneously, which allows the system to process transactions at a higher throughput. However, with multiple chains, the consensus protocol has to be enhanced to ensure the integrity and consistency of data. Ælf is a multi-chain parallel computing blockchain framework, which introduces the concept of main chain and multi-layer side chains to handle various commercial scenarios and addresses the performance bottleneck of one single chain [73]. Kan et al. proposed storing the blocks in a graph data structure instead of the the original chain and improved the performance by parallel mining [40]. Toan et al. made an alternative approach by proposing a semi-centralized network of parallel blockchain and consensus mechanism of authorized proof of stake, which aims to solve performance inefficiency of current blockchain [74]. Fitz et al. [75] presented a formal execution model that addresses the blockchain throughput and the security together. The work in [75] proposes parallel-chains, a composition technique for blockchain protocols, which achieves optimal throughput under adaptive fail-stop corruptions and is Byzantine failure tolerant.

5.1.2. Performance Benchmark and Evaluation for Verification

Although there are many new solutions proposed to solve the technical challenges of blockchain, there are few blockchain benchmark suites available to test the performance of these solutions. To provide an environment for comparing these solutions, a more comprehensive and general performance benchmark suite is urgently required. In developing such a general benchmark suite, several requirements need to meet [31]. First, the benchmark suite should be open-source. Currently, some evaluation works do not provide the source code, thus the same evaluation on different platforms cannot be repeated and validated. Secondly, the standards of performance metrics should be defined; otherwise, the evaluation results are hard to be aligned. Thirdly, common benchmark use cases should be discussed and defined. With proper use cases, the performance evaluation results can be compared in a wide spectrum of other data storage and processing platforms including traditional databases.

*5.2. Summary of Security Limitations and Possible Directions*

First, as discussed above, blockchain can be overloaded by DDoS attack, although blockchain has been viewed as a potential technique to fight DDoS . DDoS can exhaust huge resources of the network, which makes legitimate users in blockchain unable to respond to service requests promptly. The target blockchain system, especially a blockchain system that is not very distributed in fact, might not be able to deliver expected services including reaching a consensus, creating a new block, etc. Therefore, the system is in fact brought down. Approaches that can enhance the security against DDoS attacks should be investigated by future research.

Secondly, cryptography is used in blockchain to ensure identity security. Nowadays, the rise of quantum computers will bring new challenges and the most popular public-key algorithms may be easily broken by a sufficiently strong hypothetical quantum computer. Thus, anti-quantum algorithms should be explored to improve the cryptograph algorithm security, including blind signature, ring signature, and aggregate signature.

Thirdly, blockchain and smart contracts are code-based and have always been the target of hackers. A more strict and comprehensive testing procedure is required for blockchain codes and smart contracts to increase the code security.

Fourthly, the Byzantine fault tolerant technique guarantees the system security only when the mining power controlled by malicious nodes are strictly less than 50%. Thus, the 51% attack is possible, if any malicious node possesses at least 51% of the overall computational power. The malicious nodes might be able to monopolize the consensus mechanism, and force the rest of the network to erase their transactions. This means attackers can maliciously use their majority power to spend coins or tokens more than once, which is called double-spending.

Finally, the privacy issues of blockchain need more investigation. Data providers often want to keep the sensitive data private, but the transactions on the public chain are open and transparent. How to protect private data with secure algorithms remains open.

*5.3. More Blockchain Based Business Models Are Expected*

Blockchain can add significant values to many applications in different industrial domains, and can enable new business models that have not been possible before. According to our survey, currently, most proposed blockchain based business models are crypto-currency related. Blockchain features such as digital identity, distributed security, consensus-type agreements, and operation without a middleman are able to enhance the competitive capabilities and flexibility of many industries. Thus, it is important to consider how new business models can be developed, deployed and measured.

**6. Conclusions**

Blockchain's native features of decentralization, enhanced security, tamper-proof, improved traceability, transparency and the potential of increasing efficiency and speed and reducing cost can

enable and support industrial applications in many domains. Its various abilities empower multiple parties to efficiently work together, including making data sharing easier between multiple parties, transferring value in a digital way and eliminating the need for a costly third party. Thus, a wide spectrum of industry sectors are adopting or considering to take advantage of this technology in hopes of streamlining processes, enhancing security and data sharing, increasing efficiency and ultimately reducing costs.

In this paper, we survey the recent progress of blockchain from the performance and security perspective. Through the study, we are able to identify the main performance and security challenges faced by blockchain. We also introduce the state-of-the-art techniques that have been used to address these challenges. Many studies have been done and many features have been explored to improve the performance and security, and balance the trade-off between the two aspects of blockchain. Nevertheless, there are still many other issues that need more efforts, and the possible research directions include parallelism, performance benchmark, DDoS protection, anti-quantum cryptography, code security, consensus protocols and privacy.

**Author Contributions:** Funding acquisition, X.S.; Investigation, X.Z.; Methodology, Y.Z.; Writing – original draft, X.Z.; Writing – review and editing, Y.Z.

**Funding:** This research was funded by the Strategic Priority Research Program of Chinese Academy of Sciences (grant No. XDA19090106), and the Shanghai Committee of Science and Technology, China under Grant 19ZR1463900.

**Conflicts of Interest:** The funders had no role in the design of the study; in the collection, analyses, or interpretation of data; in the writing of the manuscript, or in the decision to publish the results.

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
