# Peer review of "A Survey on Challenges and Progresses in Blockchain Technologies: A Performance and Security Perspective"

_applsci, doi:10.3390/app9224731_

Round 1

Reviewer 1 Report

Title: 

A survey on challenges and progresses in blockchain

technologies: performance and security perspectives

Improve the Abstract to present the instruments (tools) used in the investigation and the method of investigation. A line should present the result(s) of the investigation. Lines 1-9 are written in the form of an introduction.  

Line 1 and line 16 are repeated sentences. 

Line 46: guarantee at 2 ... change to guarantee 2

Author Response

RE: applsci-618331

Title: A survey on challenges and progresses in blockchain technologies: a performance and security perspective

Manuscript Type: Regular

Dear Editors and Reviewers,

On behalf of my co-authors, I would like to submit as the contact author the revision of our paper ”applsci-618331” titled ”A survey on challenges and progresses in blockchain technologies: a performance and security perspective” to the special issue on ”Advances in Blockchain Technology and Applications” of your distinguished journal ”Applied Science”.

We have made extensive revision to our paper in response to the review comments.

The following is a summary of major changes

All comments from the reviewers have been addressed. We surveyed a new paper which evaluates the scalability performance of Hyperledger Fabric (Section 3.4). New paragraphs/sentences have been added in response to individual review comments. New paragraphs or text with signifificant changes are colored in red.

Please refer to the attachment for details in PDF format.

We very much appreciate your effort in reviewing our paper. We hope our paper fits your prestigious journal again as our previous publication. Looking forward to your valuable comments and guidance.

Best Regards,

Yongxin Zhu

Reviewer 2 Report

The paper presents the state-of-the-art progress of blockchain from the performance and security perspective.

The paper is generally well structured and presents interesting research, but need to be improved in the following aspects:
- The performance discussion should be somehow corroborated by experiments or measurement data.
- The application section is somehow questionable - please the authors should elaborate on why it is needed.

Author Response

*************************************************************

Reviewer: 2

Comment:

Improve the Abstract to present the instruments (tools) used in the investigation and the method of investigation.

A line should present the result(s) of the investigation. Lines 1-9 are written in the form of an introduction. Line

1 and line 16 are repeated sentences. Line 46: guarantee at 2 ... change to guarantee 2

Response:

Many thanks for your kind suggestion. We fixed the typo and rephrased the abstract.

”Blockchain naturally fifits multiple industry sectors due its natures of decentralization, enhanced security, tamper proof, improved traceability and transparency. However, there is a signifificant concern of blockchain’s performance, since blockchain tradeoffs its performance for a completely distributed feature which enhances its security. In this paper, we investigate the state-of-the-art progress of blockchain, mainly from the performance and security perspectives. We have extracted 42 primary papers from major scientifific databases and 34 online technical articles. The objective is to understand the current research trends, challenges and future directions. We brieflfly introduce the key technologies of blockchain including distributed ledgers, cryptography, consensus, smart contracts and benchmarks. We next summarize the performance and security concerns raised in the investigation. We discuss the architectural choices, performance metrics, database management enhancements, and hybrid blockchains, and try to identify the effort that the state-of-the-art made to balance between the performance and security. We also make experiments on Ethereum and survey other popular blockchain platforms on the scalability feature of blockchain. We later discuss the potential applications and present the lessons we learned. Finally we attempt to identify the open issues and possible research directions. ”

This manuscript is a resubmission of an earlier submission. The following is a list of the peer review reports and author responses from that submission.

Round 1

Reviewer 1 Report

This paper presents an overview of the key technologies of blockchain and study the current development in the performance and security aspects. Unfortunately, the reviewer does not see any value-add of this paper to the literature. Most of the findings in Section 3 are either intuitive or are already well-established. The authors included a lot of information in the paper but did not provide their own qualitative nor quantitative evaluation of these techniques or applications. The undesirable result of this is that the paper reads like a summary after what seems to be an extensive online search of information. 

The comparison tables should also be improved. For example in Table 2, it should be clearer if "Blockchain" is referring to public or private blockchain; a private blockchain may possess some characteristics that resemble a DBMS. It is also unclear the meaning of digital signature as admission control; a public blockchain can be joined by anyone but a private blockchain cannot. Another example in Table 3, where the reviewer finds the comparison in the aspect of "security" is not at the same level. If PoW and PoS are for public blockchain, then PBFT should be for Private/Federated blockchain. The phrase "verified participants" is also synonymous to "known identity". Thus there are rooms for improvement here.

There are also some errors which are potentially misleading. For example in line 117, it should be noted that Ethereum has not yet moved to PoS. The writing from line 117 to line 125 seems to suggest that BitShares improves the PoS consensus of Ethereum, which is not true. And it remains to be verified if DPoS is a better solution to PoS. Another factual error is in line 481, where Lightning Network is unlikely an option for inter-blockchain transactions. 

Overall, the paper needs significant improvement, where the authors should provide their own qualitative or quantitative evaluation in terms of performance and security instead of purely summarising existing techniques.

Reviewer 2 Report

revisit the following lines.

line 94: thus,

line 394: Thus, ...

Note: throughout the work, thus has been used without a comma.

line 655:  recast to - faced by blockchain or facing blockchain

line 656: There are a lot of work have been done ......

Reviewer 3 Report

The paper presents state-of-the-art progress of blockchain from the performance and security perspective.

The paper is generally well structured and presents interesting research, but need to be improved in the following aspects:

The criteria for choosing Ethereum, Hyperledge and Parity need to be justified. Maybe other popular or emerging platforms could be included (e.g. IOTA).

The title and abstract focus on performance and security, but performance is too briefly analyzed and discussed.

The background section should maybe also elaborate on performance and security fundaments of blockchain. 

The performance discussion regarding transactions per second and comparison to e.g. VISA in trivial and not really a contribution. 

Additionally, the performance discussion should be somehow corroborated by experiments or measurement data.

The application section is somehow questionable - please the authors should elaborate why it is needed.